# The Relationship between Discrepancies in Career Anchors of Information Technology Personnel and Career Satisfaction

**DOI:** 10.3390/bs13090785

**Published:** 2023-09-21

**Authors:** Ling-Hsing Chang, Sheng Wu

**Affiliations:** 1Department of Information Management, National Sun Yat-Sen University, Kaohsiung 80424, Taiwan; cchangmis@gmail.com; 2Department of Information Management, Southern Taiwan University of Science and Technology, Tainan 71005, Taiwan

**Keywords:** career anchors, internal career desires, external career opportunities, discrepancy theory, career satisfaction

## Abstract

The career anchors of information technology personnel (ITP) are critical factors influencing their career satisfaction (CS), and these factors are also influenced by national culture. Although a number of scholars have studied the internal CS of employees, these scholars have not explained how to increase the CS of ITP from both individual and organizational perspectives and to further improve the success rate of IS projects. Thus, this study adopts the goal–achievement gap (discrepancy) theory to explore the gap between the “internal career desires (career wants, CW)” and “external career opportunities (career have, CH)” of ITP in two different cultural societies, namely mainland China and the United Arab Emirates, and whether the gap impacts their CS. The data in this study were collected from the Internet. A survey was posted on Internet discussion forums for full-time ITP participants within organizations in China and the United Arab Emirates; thus, the results of this study are possibly only generalizable to these two countries. Finally, the results of this study provide the following contributions: (1) There are 13 career anchors (technical competence, managerial competence, autonomy, organizational stability, challenge, lifestyle, identity, creativity, variety, service, entrepreneurship, geographic security, and learning motivation) of ITP in China, which can be divided into three categories, and these are totally different from the four categories identified by ITP in the United Arab Emirates. (2) The surface analysis approach (RSA) to test the curvilinear relationship between the CW, CH, and CS of ITP indeed can explain more than the linear SEM (structural equation modeling) test between the CW and CS, CH, and CS separately, both tests are in two different cultural societies, China and the United Arab Emirates.

## 1. Introduction

Since 2000, the rapid development of information technology (IT), the Internet, information systems (IS), social network services (SNSs), and artificial intelligence (AI) have indeed been significant issues to support enterprises in offering their products and services to their global customers. In addition, IT was also pushed to develop and to progress from the SARS crisis in 2003, to the COVID-19 (Coronavirus 2019) in 2019, which broke out globally in the spring of 2020. Thus, IT has become an essential tool for a range of purposes, apart from some physical transactions and tasks (e.g., e-commerce, e-learning, telemedicine, online shopping) [1,2,3,4]. In light of this fact, information technology personnel (ITP) are more affected by newly developed IT [5,6], and the role of ITP will be much more critical in the future. Therefore, organizations will need to retain qualified ITP to improve their core capabilities because qualified information systems (IS) will be a useful asset to increase the core capability of organizations, and they need to be maintained well by qualified ITP [6]. However, ITP always have a high desire to extend their knowledge beyond the IT/IS field [6,7]. If they do not have much chance to learn the state of the art, they will leave the organization sooner or later [5,8]. Meanwhile, organizations have to train new ITP to be familiar with their business processes, and it is difficult to sustain their productivity and the quality of IS. Therefore, a high turnover rate of ITP is costly. Recently, both professionals and academics have studied how to retain qualified ITP from a “career anchor” perspective [9,10].

An effective professional has to be equipped to cope with any problem so they can accomplish unique goals with limited resources within critical time constraints [11,12]. For this reason, ITP has to receive resource allocation from IS project managers to lead IS projects and take action to recover if the project does not make the desired progress [13,14]. Therefore, ITP are the key stakeholders in the IS project management process [15,16]. Because ITP have to keep moving from project to project to develop their careers, the processes linking their careers should be critically examined [17]. The objective of ITP career development is to improve the success rate of IS projects by offering the possibility of gaining knowledge and experience [18]. The establishment of ITP in an organization promotes the professionalism and the competence of ITP management [19]. Therefore, as long as the inducing factors are found and applied to the leading turnover factors of ITP, the turnover rate of ITP and the costs of the organization will be reduced.

Although a number of scholars have studied the internal career satisfaction of employees and the important skills of ITP, ITP still need to enhance their skills via their career path [10,20], aspirations, and career success [19,20]. However, those studies did not explain how to increase the CS of ITP from either an individual or an organizational perspective and, thus, further improve the success rate of IS projects. Therefore, the “external career opportunities (career have, CH)” of an organization should match the “internal career desires (career wants, CW)” of ITP to satisfy their career desires, and this is a critical issue to retain them. For this reason, Michalos’s goal–achievement gap (discrepancy) theory [21,22] can achieve this goal and increase the CS of ITP, thereby resolving the problem of high turnover rate among ITP.

To improve the performance and the success rate of IS projects, increasing CS and reducing the turnover intentions of ITP is a feasible strategy [23,24,25]. However, from the perspective of the goal–achievement gap (discrepancy) in career anchor studies, an understanding of how to satisfy the career anchors of ITP is still lacking. There also is a shortage of research aimed at understanding the gap between the CW and CH of ITP and how the gap (between CW and CH) further affects their CS. Given that it is necessary for organizations to treat these ITP as valuable human resources, organizations have to be willing to take steps to retain personnel [12,17]. To fill the research gap, this paper is expected to help employers better attract, motivate, and retain valuable ITP for the productivity and quality of IS projects. The current study tries to figure out the nature of the gap (discrepancy) between CW and CH of ITP and whether this gap increases their CS. This is the first research question of the current study.

Because culture is another important factor influencing the career anchors of ITP, previous studies have established CW only [10,20], and scholars have not compared the gap between CW and CH from different cultural areas. In addition, China and East Asia (including Chinese society) have long been influenced by Confucianism, while the Middle East and Southeast Asia are influenced by Islam; these two areas have different religions and developed different cultures among their people. Therefore, this study has collected IT/IS personnel data from these two areas and expects to compare the finding of different kinds of relationship between ITP career anchor gaps (CW and CH) and CS. This is the second research question this study will resolve.

Finally, via the survey method of collecting ITP data, the contribution of the current study can clarify the gap between CW and CH in two different cultural contexts and the pattern’s impacts on CS. The result of the study will provide insight into the need for organizations to retain valuable ITP to increase the quality of IS projects in these two different societies and provide research model for different two cultural areas based on a survey method. However, the prime limitation of this study has not collected qualitative data to prove why there are the different results in China and the United Arab Emirates. At the same time, because of the limitation of budget and time, this study does not collect career anchors’ data of ITP from other cultural societies, and compares the difference of the result between different cultural societies. In addition, it is necessary to collect more data to provide sound hypotheses to scholars, and it will be a valuable research direction in future.

## 2. Theoretical Backgrounds

### 2.1. Career Anchor Theory

Schein [26] provided the concept of career anchor first, which is the career guidance for employees. Since a career is an experience and also a sequence of work-related positions for each individual in their lifetime, they will manage their career to align with their goals [27]. Therefore, Schein [26] defined career anchors as an individual’s self-perception of his/her talents and abilities, needs and motives, and attitude and values. Based on the study of Schein [26] and Delong [28], many scholars have studied the career anchors of ITP [24,29,30,31,32]. There are thirteen anchors as follows: technical competence, managerial competence, autonomy, organizational stability, challenge, lifestyle, identity, creativity, variety, service, entrepreneurship, geographic security, and learning motivation. For the definition of each career anchor, please refer to Appendix A [27].

Because ITP have different career needs, organizations should attempt to provide incentives that are consistent with underlying career values [10]. Therefore, some additional characteristics of the career anchor of the ITP are as follows: (1) Managerial competence anchor, i.e., ITP have to combine this competence with technical competence to perform qualified IS; thus, this anchor is a critical skill for them [9]. (2) Technological competence anchor: ITP have to keep improving their IT knowledge, skills, and abilities to the state-of-the-art; therefore, this anchor is critical to ITP [6]. Meanwhile, ITP will change their career anchors through their career stages if they have advanced greatly in their technological competence [6]. In addition, both technical competence and security will be critical to ITP at all their career stages, but managerial competence is more important at their latter career stage [6]. There is a positive relationship between the CS of the ITP and the service anchor [32]. Although many scholars have studied the anchors’ issue of ITP, most of these researches only focus on the career anchors that ITP need (CW), but they do not focus on CH. Particularly, scholars did not explore the gap between CW and CH. Thus, this study reviews the discrepancy theory as follows.

### 2.2. Discrepancy Theory

Porter [33] has found that the management can perceive that the need satisfaction of employees is an important variance for measuring the level of management and figured out that the highest-order need of self-actualization is the most critical, and it exhibits the perceived deficiency in fulfillment and perceived importance to the individual in both bottom and middle management. Then, it is the initiation of the discrepancy theory. The goal–achievement gap (discrepancy) theory is provided by Lavallee and Hatch [34], and Michalos [21] asserted that the greater the discrepancy is, the larger the gap is between people’s haves and desires. If they have a lower level of life satisfaction, then they want to achieve their desire. Therefore, the discrepancies between one’s current life and one’s desired life become the best predictors of one’s life satisfaction and also the strongest predictors in many dimensions of life satisfaction (e.g., satisfaction with friendships, family relations, housing, and even spiritual fulfillment) [34].

Over the last 60 years, several industrial/organizational psychologists have proposed discrepancy theories of pay and job satisfaction, such as the pay satisfaction integrative model, as well as positive and negative discrepancies depending on the specific combination of dissatisfaction and satisfaction [21,23,35,36]. The discrepancy theory is the explanation of the determinants of ITP’s CS [24].

Because discrepancy theory suggests that individual satisfaction is determined by the gap between actual rewards or performance and the individual’s goals or expectations, the higher initial expectations would later be less satisfied, and there would be a greater expectation–performance gap for entrepreneurs [24,37]. Because job satisfaction is principally determined by the appraisal of job experiences against “job wants”, management focused on want-have discrepancies (gap) (e.g., the gap between “what workers have been supported by their organization” and “what they want”) can predict satisfaction [21,23,24,35,36,38]. In addition, goal–achievement gap can explore the cause of the stress problem of employees and resolve the gap between “person wants” and “workplace have” problems, and it can predict job satisfaction, organizational commitment, intention to leave, job survival, and job performance for newcomers [39].

The discrepancy theory figures out that the discrepancy is the result of a psychological process of comparison between an individual’s current job experiences and their desired level, and the result will influence an individual’s job satisfaction. (e.g., what workers want now and what they have experienced in the past) [38,40]. Meanwhile, there are positive and negative discrepancies, and the result comparison depends on the individual’s emphasized level of and their job facet [24]. Therefore, this theory can predict satisfaction well [21,25].

In addition, some factors of needs of the employees (such as promotion opportunity, decision-making, freedom to perform tasks in own way, opportunity to learn new things, status, security, and skills) must be taken into account when evaluating the constructs of satisfaction and motivation, and these factors have the similar meanings with some career anchors (such as managerial competence, autonomy, organizational security, and technical competence) [24]. They extend the discrepancy theory to figure out the discrepancy between the career wants of ITP and the career haves their organization provided and perceived by ITP; then, the gap is closely related to the CS indicators, which, in turn, influences their turnover intention [24]. It acknowledges that CS is related to the extent to which job outcome (e.g., pay, reward, promotion) matches an individual’s desires. The closer the match between individual wants and organizational haves, the higher the CS and the lower the turnover intention [23].

CW is an individual’s self-concept and career values, and they will not give it up in any situation [26,41]. CH is provided by an organization and supports their CW; it includes intangible things, such as location and challenge [24,25]. Because ITP have different career needs, organizations should attempt to provide incentives that are consistent with underlying career values [25,37]. Therefore, organizations should support external opportunities (CH) to match internal anchors’ needs (CW), and it can affect the CS of ITP [24]. Scholars have found that if an organization supports and facilitates an individual’s CW appropriately at any level, their CS will be increased and the turnover rate will be decreased [24,25]. Therefore, ITP and their organization share a joint responsibility in effectively planning a career path that suits the ITP and reduces their turnover intention [23]. The organization allowing the growth of ITP should create opportunities to keep elite employees satisfied and motivated [42].

Although a number of scholars have studied the internal career and the important skills of ITP [9,20,27], valuable ITP should enhance their skills through their career path, career aspirations, and career success of IS project [24]. They have studied the ITP turnover with a discrepancy model. However, those studies did not explore the kind of relationship between the discrepancy of career anchors (the gap between CW and CH) and the CS of ITP and further improve the success rate of IS projects [24]. For this reason, it is worth exploring this relationship so that organizations can manage ITP with a more effective approach. Then, the goal–achievement gap (discrepancy) theory of Michalos [21,22] cannot only explore the above question of this study to understand how to increase the success rate of IS projects and the CS of ITP but can also resolve the problem of high turnover rate of ITP.

### 2.3. Career Anchors and Culture

Because of cultural impact, one’s values and needs will be shaped by career anchors [37,38,43,44]. Therefore, one’s workplace will influence the value of one’s career anchors [36,43], and it is necessary to investigate the career anchors of ITP from different cultures [45,46]. For this reason, the studies of Chang, Shen [20], Huang, and Aaltio [47] have found that culture indeed has a significant effect on the career anchor of the ITP. Meanwhile, an individual’s CW will also be shaped by culture [36,48].

Based on a survey study, Chang and Shen [20] explored career anchors and job satisfaction among the ITP in five different cultural areas (Mainland China, Taiwan, India, the United Arab Emirates, and the USA). Their report exhibited that the *guanxi* (*Guanxi*: literally means *relationships* and stands for any type of relationship. In the Chinese business world, however, it is also understood as the network of relationships among various parties that cooperate together and support one another. A Chinese businessman’s mentality is very much one of “You scratch my back, I’ll scratch yours”. In essence, this boils down to exchanging favors, which are expected to be performed regularly and voluntarily. Therefore, it is an important concept to understand if one is to function effectively in Chinese society) culture has a different effect on each career anchor, and each one also has a different effect on job satisfaction in the five different cultural societies. Consequently, the culture has a significant effect on the career anchor of the ITP as their needs and wants are shaped by specific cultures [36].

However, most of the cross-cultural existing empirical research is conducted in CW of ITP [20,27] and CS [24]. A dearth of research includes the CH of ITP and collects data from different cultural areas to explore the relationship between the discrepancy of career anchors (the gap between CW and CH) and the CS of ITP [24,25]. Nowadays, the discrepancy perspectives have still been ignored, and there is no research in this field. As a matter of fact, in order to increase the capability of the manpower of the IT/IS industry, it is very critical to understand the characteristics of the gap between the career anchors of CW and CH of ITP from different cultural contexts. For this reason, this study collects data from different cultural areas to understand the relationship between the career anchors gap (between CW and CH) and CS.

## 3. Methodology

### 3.1. Survey Administration

This study has collected data by surveying the Internet. A survey post was announced on the Internet discussion forum addressed to members of the Chinese and the United Arab Emirates societies to recruit organizational full-time ITP participants. This study will announce that only ITP are eligible to participate in the survey; all participants’ responses will be anonymous to ensure their confidentiality for academic purposes only. All respondents consented to use survey to collect data, and the one-year survey was conducted.

There were 369 responses collected from China (202) and United Arab Emirates (167) in this study; the valid response rate of China is 88.61% and 78.44% for the the United Arab Emirates. The valid survey number is 179 in China and 131 in the United Arab Emirates. The sample demographics information of these participants is listed in Table 1, and most participants were under the age of 40 and with Bachelor’s or Master’s degrees. The participants have similar sample structure to the studies of Armstrong, Brooks [5], Fu, and Chen [49]. Since the samples in this study include various industries, various age groups, higher education levels, different positions, work experience, and marital status, the representativeness of the sample in this study was assured.

### 3.2. Measurement Development

The measure items of this study have adopted prior validated studies, and some statement of these items was modified to fit the current study. There are 14 constructs in this study and 61 items in the questionnaire (see Appendix B). Moreover, 56 measure items of career anchors adopted from the studies of DeLong [28], Igbaria and Baroudi [50], and Chang and Lin [51] were used to measure the career anchors. Also, 5 measure items of CS were adopted from Greenhaus, Parasuraman [52]. Thus, the survey of this study has 61 questions items, each of 14 constructs has at least three indicators to be measured, all items use 5-point Likert scale (from 1 (strongly disagree) to 5 (strongly agree)) and set the direction of causality from indicator to construct.

English is the second official language in the United Arab Emirates; therefore, it is used in this survey in the United Arab Emirates. The Simplified Chinese is the official language in China; thus, the questionnaire items were translated into Simplified Chinese. Two professional translators independently translated the English questions into Chinese version, and furthermore it is reviewed by one professor to ensure the consistency of meaning between two languages. Then, the Chinese questions were translated back into English version by the other professional translator to identify their consistency. Finally, it is critical for two translators collectively to compare the original English version with back English version, and confirm the originality of these survey question. The experts of IT/IS have reviewed the questionnaire to ensure the validity, clarify the ambiguity, and modify some statement of questions.

### 3.3. Reliability and Validity

The PLS can measure research model and meanwhile evaluate the item reliability, convergent validity, and discriminant validity. The acceptable level of indicator reliability is always 0.6~0.7 [53]. The acceptable internal consistency of all scales is (CR > 0.6; AVE > 0.5) [53]. Composite reliability (CR > 0.6) and factor loading (>0.5) can ensure the reliability. The results of this study indicated that all CRs exceeded 0.6 (range: 0.72~0.88), the factor loading of all indicators exceeded 0.5 (range: 0.53~0.98), and all criteria met the standard of reliability (see Table 2). Convergent validity is necessary to measure that one construct has multiple indicators. The CR (> 0.6) and each construct’s average variance extracted (AVE > 0.5) can be tested by convergent validity [53]. The result of this study indicated that CRs exceeded 0.6 (range: 0.72~0.88), and all AVEs exceeded 0.5 (range: 0.53~0.78). Discriminant validity and the correlation between construct pairs should be lower than 0.90, and the square root of the AVE should be higher than the inter-construct correlation coefficients [53]; the data of the current study indicates that all the minimum requirements were met (see Appendix C, Table A1, Table A2, Table A3 and Table A4).

Because of the ability to validate multiple causal relationships simultaneously, PLS was used in this study, and it is an appropriate analytical tool to analyze the data of the current study. The smart PLS 3 has bootstrapping as a resampling technique (5000 random samples) to estimate the structural model and the significance of the paths (t-value) [53,54]. The structural model has been examined to describe the relationships or paths among the theoretical constructs. Each construct is a related measurement, which links the construct in the diagram to a set of items. Thus, the PLS recognizes two components of model building as follows: measurement model and structural model.

The researcher assessed the measurement model first, then tested the significant consistency in the relationship between the constructs and measurement items and between the convergent and discriminant validity (see Appendix C, Table A1, Table A2, Table A3 and Table A4). The structural model assesses the explanatory power of the independent variables and examines the size and significance of the path coefficients. Finally, the measurement and structural models form a network of measures and constructs [53].

The path significance of the relationship between CW and CS and the relationship between CH and CS of ITP in Chinese society are evaluated to explore whether the linear relationship can help management to increase the CS of ITP and the variance (R^2^ value) to explain each examined path. This study assessed the t-statistics of the path coefficients and calculated the *p*-values based on a 2-tailed test with a significance level of 0.001~0.1 firstly, then evaluated the significance and the relative strength of individual paths (see Table 3). Finally, each VIF is lower than three independent variables (see Appendix C, Table A1, Table A2, Table A3 and Table A4), and there is no collinearity problem in the research model.

This study considers nonlinear relationships in the congruence (or discrepancy) between CW and CH, two independent variables, and the relationship between CW, CH, and the dependent variable CS. Although the difference scores (discrepancy) is one of the analyzed approaches, it has important problems and reduction in information in the analyzed process [25]. Therefore, this study has applied polynomial regression analysis to test discrepancies and avoid problems caused by the difference scores [55].

There are two major principles of the polynomial regression approach: (1) congruence as a single score and as the correspondence between the measurements in a two-dimensional (2-D) space. Therefore, congruence as a line of equality can capture the absolute levels of the components and reveal the magnitude and direction of incongruence. Meanwhile, the magnitude of difference scores of the components can be ignored. (2) The effect of congruence (or discrepancy) on an outcome should be treated as a three-dimensional (3-D) surface relating the two components (CW, CH) to the outcome (CS). For this reason, this approach provides the use of difference scores as well as nonlinear relationships, and it is more complex than difference scores [55].

Therefore, in the response surface analysis approach (RSA), polynomial regression is conducted first [56]. The general form of the equation to test for relationships using polynomial regression is Z = b_0_ + b_1_ × X + b_2_ × Y + b_3_ × X^2^ + b_4_ × X × Y + b_5_ × Y^2^ + e. The predictor variables are X (internal career desires, career wants, CW), Y (external career opportunities, career have, CH); Dependent variable: Z (CS); b_1_ is beta coefficient for X; b_2_ is beta coefficient for Y; b_3_ is beta coefficient for X squared; b_4_ is cross-product of X and Y; b_5_ is beta coefficient for Y squared. When surfaces are curvilinear, the surface can be one of several types, including a concave or convex U-shape or an asymptotic form where the function approaches a limit [56].

Thus, it can analyze the degree of discrepancy between CW, CH, and CS of each ITP’s career anchor (curvilinear, U-shaped relationship) (see Figure 1 and Figure 2 and Table 4).

In essence, RSA replaces a difference score with the component measures in a nonlinear regression. Interactions and higher-order terms are included. This approach provides comprehensive tests of relationships and examines complexities that linear modeling cannot represent [55,56]. In RSA, congruence is not viewed as a single score but instead as the correspondence between the component measures (two predictors). A perfect congruence is not a point but a curve along which component measures are equal. Thus, RSA allows relationships that are more complex than simple difference scores or even linear models of the component scores. However, the correct interpretation of RSA results presupposes that the parameters of RSA cannot be interpreted in isolation and that standard RSA cannot detect significant consistency effects in the mismatch direction [57].

Polynomial Regression with Response Surface Analysis (PRRSA) [56]:Predictor variables: X (internal career desires, career wants, CW),Y (external career opportunities, career have, CH);Dependent variable: Z (career satisfaction, CS).

Z = b_0_ + b_1_ × X + b_2_ × Y + b_3_ × X^2^ + b_4_ × X × Y + b_5_ × Y^2^ + e

b_1_ is beta coefficient for X; b_2_ is beta coefficient for Y; b_3_ is beta coefficient for X squared; b_4_ is cross-product of X and Y; b_5_ is beta coefficient for Y squared.

➢a_1_: Slope along X = Y (as related to Z); b_1_ + b_2_; The value a_1_ corresponds to the slope of the surface along the line of perfect agreement between two predictor variables (X = Y) where it is related to the dependent variable Z.

Positive slope (significant positive a_1_ value): when X and Y were in agreement, Z increased as X and Y increased.

Negative slope (significant negative a_1_ value): when X and Y were in agreement, Z decreased as X and Y increased.

➢a_2_: Curvature on X = Y (as related to Z); b_3_ + b_4_ + b_5_; The value a_2_ corresponds to the curvature along the line of perfect agreement between two predictor variables (X = Y) as related to the dependent variable Z.

Convex surface (upward curving): a_2_ had been significant and positive, it would have suggested that the line of perfect agreement as it relates to Z is positive (Z increased as both X and Y become lower or higher from some point).

Concave surface (downward curving): a_2_ had been significant and negative, it would have suggested that the line of perfect agreement as it relates to Z is negative (Z decreased as both X and Y become lower or higher from some point).

➢a_3_: Slope along X = −Y (as related to Z); b_1_ − b_2_; The value a_3_ corresponds to the slope of the line of incongruence between two predictor variables (X = −Y) as related to the dependent variable Z.

Positive slop: A significant positive a_3_ indicates that Z is higher when the discrepancy is such that X is higher than Y.

Negative slop: A significant negative a_3_ indicates that Z is lower when the discrepancy is such that X is higher than Y.

➢a_4_: Curvature on X = −Y (as related to Z); b_3_ − b_4_ + b_5_**;** The value a_4_ corresponds to the curvature along the line of disagreement between two predictor variables (X = −Y) as related to the dependent variable Z.

Convex surface: A significant positive a_4_, indicating that as the discrepancy between X and Y increased, Z increased.

Concave surface: A significant negative a_4_, indicating that as the discrepancy between X and Y increased, Z decreased.

## 4. Result and Discussion

### 4.1. The Linear SEM Test of CW and CS, CH and CS

The four CWs of ITP, challenge, identity, service, and geographic security, have a significantly positive effect on CS in China, but only the autonomy CW of ITP has a significantly positive effect on CS in the United Arab Emirates (see Table 3). Meanwhile, three career anchors, lifestyle, variety, and geographic security, have been provided by the organization (CH) of ITP and have a significantly positive effect on CS in China, but only the technical competence anchor has been provided by the organization (CH) of ITP and has a significantly positive effect on CS in the United Arab Emirates (see Table 3).

The variances explanatory power (R^2^) of CS has achieved 52% of the ITP CW of China (see Appendix C, Table A1) and 55% of the ITP CH of China (see Appendix C, Table A2). However, the variances explanatory power (R^2^) of CS only has achieved 26.3% of the United Arab Emirates ITP CW (see Appendix C, Table A3) and 19% of the United Arab Emirates ITP CH (see Appendix C, Table A4). Because the result is that separate analysis of ITP’s CW and CH with linear cannot explain their CS well. Therefore, this study has analyzed the curvilinear (U-shaped) relationship between CH, CW, and CS of each ITP’s career anchor.

### 4.2. The Discrepancy between CW, CH, and CS (Curvilinear Relationship)

ITP in China: there are ten anchors (technical competence (a_1_ ***, −a_2_ *, −a_3_ *, −a_4_ *) (***: *p* < 0.001 (t > 3.29); **: *p* < 0.01 (t > 2.58); *: *p* < 0.05 (t > 1.96)), managerial competence (a_1_, a_2_ *, −a_3_, −a_4_), autonomy (a_1_ ***, a_2_, a_3_, −a_4_ ***), organizational stability (a_1_ ***, −a_2_, a_3_, −a_4_ *), creativity (a_1_ ***, −a_2_ *, −a_3_, −a_4_ *), variety (a_1_ ***, −a_2_, a_3_, −a_4_ ***), service (a_1_ ***, −a_2_ *, −a_3_, −a_4_ *), entrepreneurship (a_1_ *, a_2_, −a_3_, −a_4_ *), geographic security (a_1_ ***, -a_2_, −a_3_, −a_4_ *), and learning motivation (a_1_ ***, −a_2_, −a_3_, −a_4_ **)) have significant curvilinear relationship between the CW, CH, and CS. There are three anchors (challenge (a_1_ ***, a_2_, −a_3_, −a_4_), lifestyle (a_1_ *, a_2_, −a_3_, −a_4_), and identity (a_1_ ***, −a_2_, a_3_, −a_4_)) have significant linear relationship between the CW, CH, and CS (see Figure 1 and Figure 2, Table 4).

ITP in the United Arab Emirates: there are four anchors (technical competence (a_1_ *, a_2_, −a_3_, a_4_ *), managerial competence (−a_1_, a_2_ **, a_3_, a_4_), entrepreneurship (a_1_, a_2_ *, a_3_, a_4_), and variety (a_1_ *, −a_2_, −a_3_ *, a_4_ *)) have significant curvilinear relationship between the CW, CH, and CS. There are four anchors (challenge (a_1_ *, −a_2_, −a_3_, a_4_), geographic security (a_1_, −a_2_, −a_3_ ***, a_4_), creativity (a_1_, −a_2_, a_3_, a_4_), and service (a_1_ **, −a_2_, −a_3_, a_4_)) have significant linear relationship between the CW, CH, and CS. However, there are five anchors (autonomy (a_1_, a_2_, a_3_, a_4_), organizational stability (a_1_, −a_2_, a_3_, a_4_), lifestyle (a_1_, −a_2_, −a_3_, a_4_), identity (a_1_, a_2_, −a_3_, a_4_), and learning motivation (a_1_, −a_2_, −a_3_, a_4_)) have no significant relationship between the CW, CH, and CS (see Figure 1 and Figure 2, Table 4).

The result of the curvilinear analysis is better than the linear SEM test in two different cultural societies. It exhibits that combined CW and CH of ITP will explore more detailed information to explain what kinds of relationship between CW and CH will influence their CS. In order to further understand the result, it is divided into seven categories in this study.

### 4.3. Seven Categories of the Discrepancy between the CW, CH, and CS

Based on each career anchor of ITP in two different cultural societies, the value of a_1_ and a_3_ is a positive/negative slope, and the value of a_2_ and a_4_ is a convex/concave surface; then, these anchors were classified into seven categories in this study. Finally, the ITP in China can be divided into three categories: Type I, Type II, and Type III on the one side; meanwhile, the ITP in the United Arab Emirates can be divided into four categories: Type IV, Type V, Type VI, and Type VII on the other side. The result has been discussed in the following sections.

Type I: technical competence (a_1_ ***, −a_2_ *, −a_3_ *, −a_4_ *), creativity (a_1_ ***, −a_2_ *, −a_3_, −a_4_ *), service (a_1_ ***, −a_2_ *, −a_3_, −a_4_ *), geographic security (a_1_ ***, −a_2_, −a_3_, −a_4_ *), and learning motivation (a_1_ ***, −a_2_, −a_3_, −a_4_ **) of ITP in the China. In which, their CW and CH have the same relationship (positive a_1_, and negative a_2_, a_3_, a_4_) with CS. Thus, it is easy to know that these five anchors can be treated with the same way to increase the CS of ITP in China.

The above results exhibit the following: (1) Technical competence anchor: four values, a_1_, a_2_, a_3_, and a_4_, are significant for the slop and concave surface. Thus, the CS of technical competence anchor will be increased if CW and CH are in agreement (a_1_), the CS of ITP will be decreased if CW is higher than CH, and the discrepancy increased between CW and CH (−a_3_), and CS of ITP will be decreased if the discrepancy increased between CW and CH (−a_4_). However, the CS will be decreased because both CW and CH are away from some point, no matter what CW and CH are decreased or increased (−a_2_). In sum, if an organization does not provide this anchor to ITP (no CH), its CS will be the lowest level.

(2) Two anchors, creativity and service: three values, a_1_, a_2_, and a_4_, are significant for the slop and concave surface, but a_3_. Thus, the CS of these two anchors will be increased if CW and CH are in agreement (a_1_), and the CS of ITP will decrease if the discrepancy increases between CW and CH (−a_4_). However, the CS will decrease because both CW and CH are away from one specific point, no matter what CW and CH are decreased or increased (−a_2_). In short, if an organization does not provide this anchor to ITP (no CH), and ITP does not emphasize this anchor (no CW) either, their CS will be the lowest level.

(3) Two anchors, geographic security and learning motivation: two values, a_1_ and a_4_, are significant for the slop and concave surface, but a_2_, a_3_. Thus, the CS of these two anchors will be increased if CW and CH are in agreement (a_1_), and the CS of ITP will be decreased if the discrepancy increases between CW and CH (−a_4_). Therefore, if an organization provides too much or does not provide this anchor (CH) to ITP, their CS will be decreased speedily or reach the lowest level.

Type II: organizational stability (a_1_ ***, −a_2_, a_3_, −a_4_ *), variety (a_1_ ***, −a_2_, a_3_, −a_4_ ***), autonomy (a_1_ ***, a_2_, a_3_, −a_4_ ***), and identity (a_1_ ***, −a_2_, a_3_, −a_4_) of ITP in the China. In which, their CW and CH of three anchors have the same relationship (positive a_1_, a_3_, and negative a_2_, a_4_) with CS. Although the a_2_ value of autonomy anchor is positive, however, all of these four anchors’ a_2_ are not significant in the relationship between the CW, CH, and CS. Thus, we classify these four anchors belonging to Type II category. In light of this, it is easy to know that these four anchors can be treated with the same way to increase the CS of ITP in China.

The above result exhibits the following: (1) Three anchors, organizational stability, variety, and autonomy: two values, a_1_ and a_4_, are significant for the slop and concave surface, but a_2_, a_3_. Thus, the CS of these two anchors will be increased if CW and CH are in agreement (a_1_), and the CS of ITP will be decreased if the discrepancy increases between CW and CH (−a_4_). In sum, if the organization provides too much or does not provide this anchor (CH) to ITP, their CS will be decreased speedily or have the lowest level.

(2) Identity anchor: only value a_1_ is significant for the slop, but a_2_, a_3_, a_4_. Thus, the CS of these two anchors will be increased if CW and CH are in agreement (a_1_). In short, if the organization does not provide this anchor to ITP (no CH), their CS will be the lowest level.

Type III: four anchors of ITP in China, entrepreneurship (a_1_ *, a_2_, −a_3_, −a_4_ *), challenge (a_1_ ***, a_2_, −a_3_, −a_4_), lifestyle (a_1_ *, a_2_, −a_3_, −a_4_), and managerial competence (a_1_, a_2_ *, −a_3_, −a_4_. Their CW and CH have the same relationship (positive a_1_, a_2_, and negative a_3_, a_4_) with CS. Thus, it is easy to know that these four anchors can be treated in the same way to increase the CS of ITP in China.

The above results exhibit the following: (1) Entrepreneurship anchor: two values, a_1_ and a_4_, are significant for the slop and concave surface, but a_2_, a_3_. Thus, the CS of these two anchors will be increased if CW and CH are in agreement (a_1_), but the CS of ITP will be decreased if the discrepancy increases between CW and CH (−a_4_). (2) Two anchors Challenge and lifestyle: only value a_1_ is significant for the slop, but a_2_, a_3_, a_4_. Thus, the CS of these two anchors will be increased if CW and CH are in agreement (a_1_). (3) Managerial competence anchor: only value a_2_ is significant for the convex surface, but a_1_, a_3_, a_4_. Thus, the CS will be increased because both CW and CH are in perfect agreement, no matter what CW and CH are decreased or increased (a_2_).

In summary, if an organization provides too much or does not provide these four anchors (CH) to ITP, their CS will be decreased speedily or have the lowest level.

Type IV: six anchors of ITP in the United Arab Emirates, variety (a_1_ *, −a_2_, −a_3_ *, a_4_ *), challenge (a_1_ *, −a_2_, −a_3_, a_4_), service (a_1_ **, −a_2_, −a_3_, a_4_), geographic security (a_1_, −a_2_, −a_3_ ***, a_4_), lifestyle (a_1_, −a_2_, −a_3_, a_4_), and learning motivation (a_1_, −a_2_, −a_3_, a_4_). Their CW and CH have the same relationship (positive a_1_, a_4_, and negative a_2_, a_3_) with CS. Thus, it is easy to know that these six anchors can be treated with the same way to increase the CS of ITP in the United Arab Emirates.

The above results exhibit the following: (1) Variety anchor: three values, a_1_, a_3_, and a_4_, are significant for the slop and convex surface. Thus, the CS of the variety anchor will be increased if CW and CH are in agreement (a_1_), and the CS of ITP will be increased when CW is higher than CH, and the discrepancy increases between CW and CH (a_4_). (2) Two anchors challenge and service: only value a_1_ is significant for the slop, but a_2_, a_3_, a_4_. Thus, the CS of these two anchors will be increased if CW and CH are in agreement (a_1_). (3) geographic security anchor: only value a_3_ is significant for the slop, but a_1_, a_2_, a_4_. Thus, the CS of ITP will be decreased if the discrepancy increases between CW and CH (−a_3_).

In short, if an organization does not provide three anchors, variety, challenge, and service, to ITP (no CH) and ITP do not emphasize this anchor (no CW) either, their CS will be the lowest level. If ITP only emphasizes geographic security anchor in a middle level, their CS will be the lowest level.

Type V: two anchors of ITP in the United Arab Emirates technical competence (a_1_ *, a_2_, −a_3_, a_4_ *) and identity (a_1_, a_2_, a_3_, a_4_). Their CW and CH have the same relationship (positive a_1_, a_2_, a_4_, and negative a_3_) with CS. Thus, it is easy to know that these two anchors can be treated in the same way to increase the CS of ITP in the United Arab Emirates.

The above results exhibit the following: (1) Technical competence anchor: two values, a_1_ and a_4_, are significant for the slop and convex surface (a_1_). Thus, the CS of the technical competence anchor will be increased if CW and CH are in agreement, and the CS of ITP will be increased if the discrepancy increases between CW and CH (a_4_). (2) Identity anchor: four values, a_1_, a_2_, a_3_, and a_4_, have no significance between the relationship of the CW, CH, and CS of ITP in the United Arab Emirates.

In summary, if an organization provides a middle-level technical competence anchor to ITP (CH), and ITP emphasizes this anchor (CW) with the middle level, too, their CS will be the lowest level.

Type VI: two anchors of them in the United Arab Emirates, creativity (a_1_ **, −a_2_, a_3_, a_4_) and organizational stability (a_1_, −a_2_, a_3_, a_4_). Their CW and CH have the same relationship (positive a_1_, a_3_, a_4_, and negative a_2_) with CS. Thus, it is easy to know that these two anchors can be treated with the same way to increase the CS of ITP in the United Arab Emirates.

The above results exhibit the following: (1) Creativity anchor: value a_1_ is significant for the slope. Thus, the CS of the technical competence anchor will be increased if CW and CH are in agreement (a_1_). (2) Organizational stability anchor: four values, a_1_, a_2_, a_3_, and a_4_, have no significance between the relationship of the CW, CH, and CS of ITP in the United Arab Emirates.

In short, if the organization provides a middle-level technical competence anchor to ITP (CH), and ITP emphasizes this anchor (CW) with the middle level, too, their CS will be the lowest level.

Type VII: three anchors in the United Arab Emirates, entrepreneurship (a_1_, a_2_ *, a_3_, a_4_), managerial competence (−a_1_, a_2_ **, a_3_, a_4_), and autonomy (a_1_, a_2_, a_3_, a_4_). In which, their CW and CH of two anchors, entrepreneurship and autonomy have the same relationship (positive a_1_, a_2_, a_3_, a_4_) with CS. Although the value a_1_ of entrepreneurship anchor is negative, however, all of these three anchors’ a_1_ are not significant in the relationship between the CW, CH, and CS. Thus, we classify these three anchors belonging to Type VII category. In light of this, it is easy to know that these three anchors can be treated with the same way to increase the CS of ITP in the United Arab Emirates.

The above results exhibit the following: (1) Two anchors, entrepreneurship and managerial competence: only value a_2_ is significant for the convex surface, but a_1_, a_3_, a_4_. Thus, the CS will be increased because both CW and CH are in perfect agreement, no matter what CW and CH are decreased or increased (a_2_). (2) Autonomy anchor: four values, a_1_, a_2_, a_3_, and a_4_, have no significance between the relationship of the CW, CH, and CS of ITP in the United Arab Emirates.

Therefore, if an organization provides middle level two anchors anchor entrepreneurship and managerial competence to ITP (CH), and ITP emphasizes these two anchors (CW) with middle level too, their CS will be the lowest level.

### 4.4. Compare the Categories’ Result of the ITP in Two Different Cultural Societies: China and the United Arab Emirates

This study found that two different cultural societies in China and the United Arab Emirates indeed have different results from the curvilinear analysis, and the ITP in two different cultural societies can be divided into different categories (see Figure 1 and Figure 2, Table 4 and Table 5).

(1)The categories’ result of the ITP in Chinese society

In Chinese society, the career anchors of ITP are classified into three categories: Type I, Type II, and Type III. In these three categories, most of four significant values are positive a_1_; negative a_2_, a_3_, and a_4_; therefore, the result exhibits that the curvilinear relationship between the CW, CH, and CS of Type I and Type II categories are concave surface. However, in Type III, the value a_2_ of managerial competence anchor is signif icantly positive, and value a_4_ of entrepreneurship anchor is significantly negative. Thus, both of these two anchors have concave and convex surface of the curvilinear relationship between the CW, CH, and CS at the same time.

Only three anchors, identity (Type II), challenge (Type III), and lifestyle (Type III) of its personnel, have had a significantly positive a_1_ value relationship between the CW, CH, and CS. The other eight anchors have a significant curvilinear relationship between the CW, CH, and CS. For this reason, the discrepancy method of career anchors can deeply explore the relationship between the CW, CH, and CS of ITP in China more useful than the study from the linear relationship between CW and CS, or the CH and CS.

(2)The categories’ result of the ITP in the United Arab Emirates society

In the United Arab Emirates society, the career anchors of ITP are classified into four categories: Type IV, Type V, Type VI, and Type VII. The result exhibits that the curvilinear relationship between the CW, CH, and CS of these three categories are convex surface; in these four types categories all of four significant values are positive a_1_, a_2_, and a_4_; negative a_3_.

Only four anchors, variety (Type IV), technical competence (Type V), entrepreneurship (Type VII), and managerial competence (Type VII), have had significantly positive value on a_2_ and a_4_ and express the convex surface. Three anchors, challenge (Type IV), service (Type IV), and creativity (Type VI), only have linear positive slop value a_1_ relationship between the CW, CH, and CS; the geographic security (Type IV) anchor also only has linear negative slop relationship between the CW, CH, and CS.

Moreover, the five anchors (lifestyle (Type IV), learning motivation (Type IV), identity (Type V), organizational stability (Type VI), and autonomy (Type VII)) still do not have a significant relationship between the CW, CH, and CS. The positive value a_2_ of two anchors autonomy (Type IV) and identity (Type V) have had a marginally significant relationship between the CW, CH, and CS; thus, these two anchors have expressed the curvilinear convex surface. The positive value a_1_ of two anchors, organizational stability (Type VI) and learning motivation (Type IV), also have had a marginally significant relationship between the CW, CH, and CS; thus, these two anchors express the linear slope shape.

(3)Why the result of the curvilinear relationship between China and the United Arab Emirates is different.

The result of totally different categories of ITP between the China and the United Arab Emirates exhibits that the discrepancy theory to analyze the CW and CH of ITP in China is more important and useful than the ITP in the United Arab Emirates.

The significant difference in ITP between China and the United Arab Emirates is probably because IT-related industries are the biggest industry in China [58]. With GDP per capita almost 80% higher than average value of OECD (Organization for Economic Cooperation and Development), the United Arab Emirates has been one of richest countries in the world in the recent 50 years, and the second largest economy to Saudi Arab is in the GCC (Gulf Cooperation Council). However, its major challenge is to convert investments and strong enabling conditions into knowledge, innovation and creative outputs [59].

Thus, although the United Arab Emirates government supports the IT industry strongly, and the IT industry has the speedy growth rate recently, but its IT industry is not so critical as that in China. In light of above situation, China and the United Arab Emirates obviously have different national industry strategies.

On the one hand, the scale of the IT industry can provide many ITP occupations; thus, how to retain valuable ITP is a critical issue for each organization in China. But, the labor market cannot provide enough ITP. Thus, for employing and retaining valuable ITP to increase the quality of IS project, satisfying their CW will be a useful and feasible approach; therefore, the discrepancy between CW and CH will be very significant to the ITP in China.

Meanwhile, in the Chinese context, relationships or connections between people is very crucial in Chinese daily life, workplace relationship, and social networking [60]. *Guanxi* is an essential socio-cultural concept among the Chinese and fills every aspect of their social and organizational life [61]. “Whom you know” is an essential part of *guanxi*, which is important in the organization to obtain resources available for particular individuals [62]. In Chinese society, socioeconomic background and family origins of individuals are important factors in interpersonal relations. Therefore, the *guanxi* is a kind of direct particularistic tie between an individual and others [63].

For this reason, to retain valuable ITP to increase the quality of IS project, Chinese organization will consider employees’ need frequently and provide them with expected career anchors (CH) to increase their CS. It will be the responsibility of enterprises to take care of their employees, especially valuable ITP, which is the core manpower of the IS project in an organization.

On the other hand, although the United Arab Emirates government supports the IT industry strongly, and it has the speedy growth rate recently, but their organizations didn’t pay attention to provide the career anchors ITP need. However, we collected and analyzed the data by discrepancy theory, we still can analyze the relationship between the CW, CH, and CS of ITP in the United Arab Emirates more useful than the study from the linear relationship between CW and CS, or the CH and CS.

The result probably is that the United Arab Emirates culture is based on Arabian culture which is influenced by the cultures of Persia, India, and East Africa, and is a part of the culture of Eastern Arabia. Meanwhile, Sharia courts have exclusive jurisdiction over family law cases and also several criminal cases. The Islamic personal status law can apply to Muslims and sometimes non-Muslims. Non-Muslim expatriates have to be liable to Sharia rulings on marriage, divorce, and child custody. Because the major challenge of the United Arab Emirates is converting investments and strong enabling conditions into knowledge, innovation, and creative outputs [59]; thus, their ITP expect their organization can provide variety, challenge, service, creativity, technical competence, entrepreneurship, and managerial competence critical anchors to them. However, probably because the organizations in the United Arab Emirates may not provide their ITP opportunities to support their need anchors (CH) (see Appendix C, Table A4), and the result of the discrepancy between CW, and CH, and the curvilinear relationship of the CW, CH, and CS is not so significant than the ITP in China.

## 5. Contributions

### 5.1. For Academic

Because rare scholars study career anchors of the ITP based on discrepancy theory and compare the disparity from different cultural societies. To fill the gap, this study collect data from China and the United Arab Emirates and has the following academic contributions.

Firstly, because the linear relationship between CW and CS of ITP in China only four career anchors (challenge, identity, service, and geographic security) have positive significant result; and the linear relationship between CH and CS of ITP in China only three career anchors (lifestyle, variety, and geographic security) have positive significant result. The linear relationship between CW and CS of ITP in the United Arab Emirates only autonomy anchor has positive significant result; and the linear relationship between CH and CS of ITP in the United Arab Emirates only has technical competence anchor and a positive significant result. However, all career anchors of ITP have a significance of the curvilinear relationship between the CW, CH, and CS in China, and seven anchors of ITP have the significance of the curvilinear relationship between the CW, CH, and CS in the United Arab Emirates. Thus, the curvilinear analysis method can indeed explain the linear relationship both in China and the United Arab Emirates.

Secondly, there are three situations in China: (1) When the ITP do not want (no CW) organizational stability (Type II), variety (Type II), and learning motivation (Type I) three anchors (CW), no matter what the organization whether provides (CH) these three anchors; then their CS has the lowest level. (2) IF organization does not provide (no CH) six anchors technical competence (Type I), creativity (Type I), service (Type I), geographic security (Type I), identity (Type II), and lifestyle (Type III) to ITP, their CS will be had the lowest level. (3) IF organization provides too much or too little four anchors (CH) autonomy (Type II), entrepreneurship (Type III), challenge (Type III), managerial competence (Type III) to ITP, their CS also will be the lowest level too.

Thirdly, there are four situations in the United Arab Emirates: (1) IF organization does not provide seven anchors (no CH) variety (Type IV), challenge (Type IV), service (Type IV), lifestyle (Type IV), learning motivation (Type IV), identity (Type V), and creativity (Type VI), their CS will be the lowest level. (2) When the ITP do not want (no CW) organizational stability (Type VI) anchor, no matter what the organization whether provides these three anchors, then their CS has the lowest level. (3) When the ITP only want (CW) the middle level of two anchors geographic security (Type IV), and autonomy (Type VII), no matter what the organization whether provides (CH) these three anchors; then their CS is the lowest level. (4) When the ITP want (CW) and the organization provides (CH) the middle level of three anchors, technical competence (Type V), entrepreneurship (Type VII), and managerial competence (Type VII), then their CS is the lowest level.

Finally, the results of this study can help scholars to understand the discrepancy perspective to analyze the CW and CH of ITP; thus, the highest and lowest CS can emerge easily. Meanwhile, the discrepancy Types between China and the United Arab Emirates can be the base to provide scholars further to study other cultural societies in future.

### 5.2. For Practice

Firstly, because the IT industry is the biggest industry in China, their organizations will provide to their ITP CH very close to their CW, the CH of three anchors identity, service and geographic security even more than their CW (see Appendix C, Table A1 and Table A2). Thus, for retaining valuable ITP to increase the quality of IS project, the Chinese enterprises will take care of them and build good *guanxi* with them to increase their CS. Thus, the ITP discrepancy career anchors patterns Type I, Type II, and Type III can provide management how to build suitable strategy to increase the CS of ITP.

Secondly, organizations in the United Arab Emirates indeed have lower level of the CH than the CW of their ITP (see Appendix C, Table A3 and Table A4), probably because their industries do not focus on the IT so much. For this reason, the government and enterprises in the United Arab Emirates should be more proactive to invest and enable beneficial conditions for knowledge, innovation and creative outputs to achieve the industrial upgrading goal. Therefore, when the IT industry is critical in the United Arab Emirates, enterprises will pay attention to the CW of ITP, and provide them career anchors (CH) which they emphasized; then, their CS will be increased too.

Thirdly, in light of the fact that management can understand how to provide the appropriate levels of different career anchors to the ITP in two cultural societies, China and the United Arab Emirates, and increase their CS to achieve the highest level and avoid their CS having the lowest level, then, the different career anchors’ Types of ITP also are more convenient to help management to retain valuable ITP for increasing the quality of IS project in the China and United Arab Emirates.

Finally, management can combine the curvilinear relationship between CW, CH, and CS of this study, the characteristics of national culture in China and the United Arab Emirates, and the national industry strategy to build appropriate police to achieve the goal to retain valuable ITP for increasing the quality of IS project. Therefore, the organization can improve the advantage of their competence more easily than before.

## 6. Conclusions

Because CS is a critical factor for ITP, this study adopts discrepancy theory to analyze the curvilinear relationship among the CW, CH, and CS in China and the United Arab Emirates, two different cultural societies. Meanwhile, the results of this study indeed have proved that the curvilinear relationship indeed can explain more than linear relationship among the CW and CS, CH and CS separately. In addition, not only national culture will influence the curvilinear result, the industry strategy of nation also will influence the result too. In light of this, this is a suitable approach to retain valuable ITP for increasing the quality of IS project. However, the result also needs qualitative data to prove why there are the different results in China and the United Arab Emirates, and this is a prime limitation of this study. For increasing the CS of global ITP, management can collect career anchors’ data of ITP from different cultural societies, and it will be an effective approach in the future. In addition, it is necessary to collect more data to provide sound hypotheses to scholars, and this also is another limitation of this study. Finally, this study has provided academic contributions and given practical suggestions to management.

## Figures and Tables

**Figure 1 behavsci-13-00785-f001:**
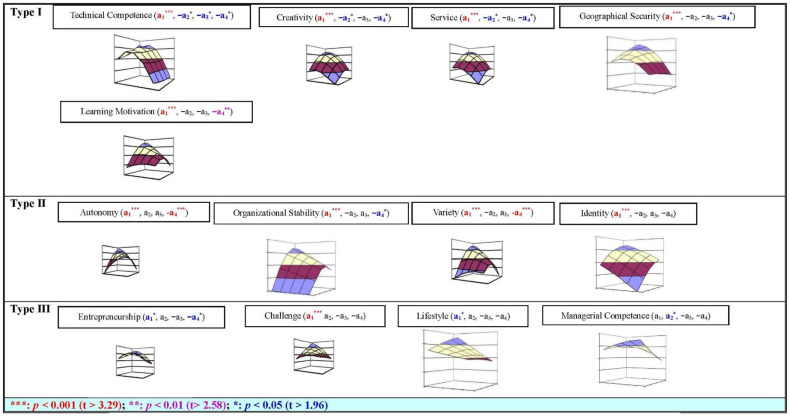
The ITP career anchors categories of curvilinear analysis result (type I~type III): China.

**Figure 2 behavsci-13-00785-f002:**
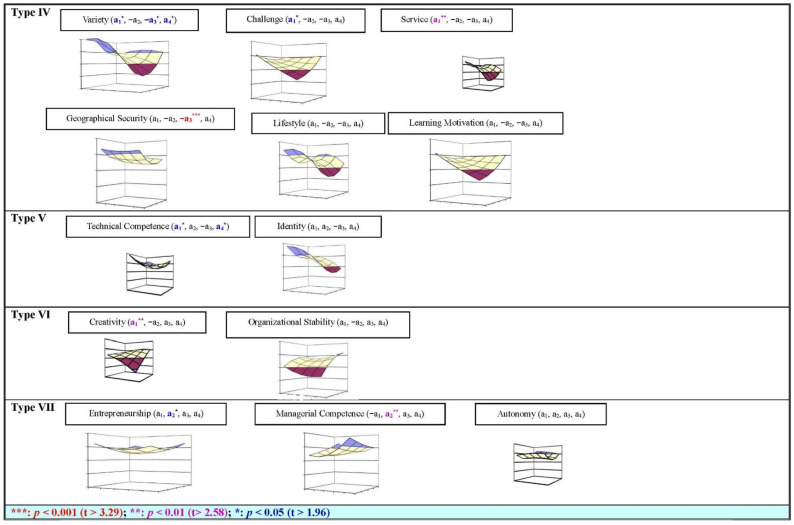
The ITP career anchors categories of curvilinear analysis result (type IV~type VII): United Arab Emirates.

**Table 1 behavsci-13-00785-t001:** Sample demographics.

Demographical Characteristics	Contents	China		United Arab Emirates	
		#	%	#	%
Gender	(1) Male	89	49.72	98	74.81
	(2) Female	90	50.28	32	24.43
	(3) Miss Value	0	0.00	1	0.76
Age	(1) ≤25	7	3.91	21	16.03
	(2) >25 and ≤30	52	29.05	43	32.82
	(3) >30 and ≤35	61	34.08	33	25.19
	(4) >35 and ≤40	55	30.73	21	16.03
	(5) >40 and ≤45	1	0.56	11	8.40
	(6) >45 and ≤50	1	0.56	2	1.53
	(7) >50 and ≤60	2	1.12	0	0.00
	(8) over 60	0	0.00	0	0.00
Education	(1) High school	1	0.56	0	0.00
	(2) Junior college graduates	1	0.56	3	2.29
	(3) Bachelor’s degree	126	70.39	87	66.42
	(4) Master	49	27.37	39	29.77
	(5) Ph.D	2	1.12	1	0.76
	(6) Miss Value	0	0.00	1	0.76
Industry	(1) Government	0	00.00	28	21.37
	(2) Information Service	114	63.69	25	19.08
	(3) Medicine	1	0.56	11	8.40
	(4) Financial	3	1.68	12	9.16
	(5) Others	61	34.07	55	41.99
Job Title	(1) System engineer	30	16.76	16	12.21
	(2) Software engineer	56	31.28	8	6.11
	(3) Hardware engineer	21	11.73	4	3.05
	(4) Network engineer	6	3.35	17	12.98
	(5) Programmer	23	12.85	8	6.11
	(6) DBA	18	10.06	2	1.53
	(7) Project manager	9	5.03	20	15.27
	(8) MIS Manager	3	1.68	7	5.34
	(9) System analyst	9	5.03	9	6.87
	(10) MIS sales	4	2.23	0	0.00
	(11) Others	0	0.00	40	30.53
Job experience	(1) 1~3 years	16	8.94	26	19.85
	(2) 4~6 years	48	26.81	28	21.37
	(3) 7~9 years	45	25.14	26	19.85
	(4) 10~13 years	42	23.46	24	18.32
	(5) 14~17 years	24	13.41	13	9.92
	(6) 18~21 years	1	0.56	6	4.58
	(7) 22 years or more	3	1.68	3	2.29
	(8) Miss Value	0	0.00	5	3.82
Married	(1) Yes	152	84.92	74	56.49
	(2) No	27	15.08	51	38.93
	(3) Miss Value	0	0.00	6	4.58

**Table 2 behavsci-13-00785-t002:** Standardized loading and reliability estimates.

Constructs	Item	China (Needs)		China (Have)		United Arab Emirates (Needs)		United Arab Emirates (Have)	
		Factor Loadings	CR (AVE)	Factor Loadings	CR (AVE)	Factor Loadings	CR (AVE)	Factor Loadings	CR (AVE)
Technical Competence	TECH1	0.85	0.79 (0.66)	0.94	0.72 (0.58)	0.82	0.83 (0.70)	0.95	0.74 (0.60)
	TECH3	0.77		0.53		0.86		0.55	
Managerial Competence	MANG2	0.69	0.80 (0.67)	0.74	0.82 (0.69)	0.60	0.78 (0.65)	0.58	0.77 (0.64)
	MANG4	0.93		0.91		0.97		0.97	
Autonomy	AUTO4	0.88	0.85 (0.74)	0.80	0.81 (0.67)	0.92	0.87 (0.78)	0.84	0.82 (0.70)
	AUTO5	0.84		0.84		0.84		0.83	
Organizational Stability	ORGS1	0.74	0.77 (0.63)	0.81	0.80 (0.67)	0.96	0.82 (0.70)	0.74	0.86 (0.76)
	ORGS2	0.84		0.82		0.69		0.98	
Challenge	CHAL2	0.71	0.80 (0.57)	0.76	0.80 (0.58)	0.63	0.76 (0.51)	0.54	0.80 (0.57)
	CHAL4	0.75		0.73		0.78		0.84	
	CHAL5	0.80		0.79		0.73		0.86	
Lifestyle	LIFE2	0.92	0.75 (0.61)	0.89	0.83 (0.71)	0.60	0.72 (0.58)	0.78	0.83 (0.71)
	LIFE4	0.61		0.80		0.89		0.91	
Identity	IDEN1	0.84	0.79 (0.65)	0.80	0.81 (0.68)	0.78	0.83 (0.72)	0.81	0.82 (0.70)
	IDEN3	0.78		0.84		0.91		0.86	
Creativity	CREA1	0.72	0.75 (0.60)	0.85	0.82 (0.69)	0.82	0.78 (0.64)	0.93	0.81 (0.68)
	CREA2	0.83		0.81		0.78		0.71	
Variety	VARI1	0.83	0.80 (0.67)	0.79	0.74 (0.59)	0.65	0.78 (0.64)	0.85	0.81 (0.68)
	VARI3	0.81		0.75		0.93		0.79	
Service	SER2	0.81	0.83 (0.70)	0.88	0.87 (0.77)	0.87	0.82 (0.69)	0.81	0.82 (0.70)
	SER3	0.87		0.87		0.79		0.86	
Entrepreneurship	ENTE2	0.87	0.82 (0.69)	0.92	0.77 (0.64)	0.86	0.86 (0.75)	0.76	0.87 (0.78)
	ENTE3	0.80		0.65		0.88		0.98	
Geographical Security	GEO2	0.71	0.78 (0.64)	0.66	0.77 (0.63)	0.74	0.85 (0.74)	0.83	0.85 (0.73)
	GEO3	0.88		0.90		0.97		0.87	
Learning Motivation	LEAR2	0.82	0.82 (0.69)	0.87	0.79 (0.65)	0.89	0.85 (0.73)	0.83	0.82 (0.69)
	LEAR3	0.84		0.74		0.82		0.83	
Career Satisfaction	CS1	0.76	0.82 (0.53)	0.78	0.82 (0.53)	0.83	0.88 (0.64)	0.72	0.87 (0.63)
	CS2	0.75		0.70		0.82		0.76	
	CS4	0.63		0.66		0.78		0.87	
	CS5	0.77		0.78		0.78		0.83	
SRMR Common Factor Model		0.08		0.08		0.08		0.09	

CR: composite reliability; AVE: average variance extracted.

**Table 3 behavsci-13-00785-t003:** Structural equation modeling test.

Hypotheses	China (CW)		China (CH)		United Arab Emirates (CW)		United Arab Emirates (CH)	
	β	t Value	β	t Value	β	t Value	β	t Value
Technical Competence → Career Satisfaction	0.073	0.933	0.074	1.112	0.145	1.774	0.225 *	2.158
Managerial Competence → Career Satisfaction	−0.019	0.339	0.042	0.602	−0.051	0.466	−0.146	1.015
Autonomy → Career Satisfaction	0.077	1.010	0.108	1.474	0.210 *	2.210	0.058	0.433
Organizational Stability → Career Satisfaction	0.033	0.569	0.101	1.477	0.055	0.562	−0.243	1.583
Challenge → Career Satisfaction	0.255 ***	3.488	0.098	1.432	−0.039	0.467	0.009	0.075
Lifestyle → Career Satisfaction	−0.065	0.951	0.159 *	2.187	0.103	1.040	0.094	0.845
Identity → Career Satisfaction	0.158 *	2.373	0.091	1.359	0.036	0.391	0.121	1.213
Creativity → Career Satisfaction	0.034	0.490	−0.043	0.659	0.147	1.359	−0.141	0.874
Variety → Career Satisfaction	0.092	1.523	0.167 *	2.492	0.176	1.531	0.084	0.739
Service → Career Satisfaction	0.185 **	2.836	0.090	1.418	0.122	1.062	0.213	1.886
Entrepreneurship → Career Satisfaction	0.090	1.688	0.029	0.514	−0.093	0.859	0.064	0.424
Geographical Security → Career Satisfaction	0.226 ***	3.761	0.153 *	2.419	−0.060	0.697	0.065	0.674
Learning Motivation → Career Satisfaction	0.021	0.316	0.080	1.128	0.038	0.435	0.022	0.208

***: *p* < 0.001 (t > 3.29); **: *p* < 0.01 (t > 2.58); *: *p* < 0.05 (t > 1.96); PLS-Bootstrap method was applied and bootstrap 5000 samples for parameters estimation. All values in table are presented by standardized regression coefficient.

**Table 4 behavsci-13-00785-t004:** The career anchors category type.

Career Anchor	China	United Arab Emirates
Technical Competence	Type I (a_1_ ***, −a_2_ *, −a_3_ *, −a_4_ *)	Type V (a_1_ *, a_2_, −a_3_, a_4_ *)
Managerial Competence	Type III (a_1_, a_2_ *, −a_3_, −a_4_)	Type VII (−a_1_, a_2_ **, a_3_, a_4_)
Autonomy	Type II (a_1_ ***, a_2_, a_3_, −a_4_ ***)	Type VII (a_1_, a_2_, a_3_, a_4_)
Organizational Stability	Type II (a_1_ ***, −a_2_, a_3_, −a_4_ *)	Type VI (a_1_, −a_2_, a_3_, a_4_)
Challenge	Type III (a_1_ ***, a_2_, −a_3_, −a_4_)	Type IV (a_1_ *, −a_2_, −a_3_, a_4_)
Lifestyle	Type III (a_1_ *, a_2_, −a_3_, −a_4_)	Type IV (a_1_, −a_2_, −a_3_, a_4_)
Identity	Type II (a_1_ ***, −a_2_, a_3_, −a_4_)	Type V (a_1_, a_2_, −a_3_, a_4_)
Creativity	Type I (a_1_ ***, −a_2_ *, −a_3_, −a_4_ *)	Type VI (a_1_ **, −a_2_, a_3_, a_4_)
Variety	Type II (a_1_ ***, −a_2_, a_3_, −a_4_ ***)	Type IV (a_1_ *, −a_2_, −a_3_ *, a_4_ *)
Service	Type I (a_1_ ***, −a_2_ *, −a_3_, −a_4_ *)	Type IV (a_1_ **, −a_2_, −a_3_, a_4_)
Entrepreneurship	Type III (a_1_ *, a_2_, −a_3_, −a_4_ *)	Type VI (a_1_, a_2_ *, a_3_, a_4_)
Geographical Security	Type I (a_1_ ***, −a_2_, -a_3_, −a_4_ *)	Type IV (a_1_, −a_2_, −a_3_ ***, a_4_)
Learning Motivation	Type I (a_1_ ***, −a_2_, −a_3_, -a_4_ **)	Type IV (a_1_, −a_2_, −a_3_, a_4_)

***: *p* < 0.001 (t > 3.29); **: *p* < 0.01 (t > 2.58); *: *p* < 0.05 (t > 1.96).

**Table 5 behavsci-13-00785-t005:** Compare the categories’ result of the ITP in two different cultural societies, China and United Arab Emirates.

China			United Arab Emirates		
Career Anchor	Result		Career Anchor	Result	
Managerial Competence	Type III (a_1_, a_2_ *, −a_3_, −a_4_)	The value a_2_ of managerial competence anchor is significantly positive, and a_4_ value a_4_ of entrepreneurship anchor is significantly negative. Both of these two anchors have concave and convex surfaces of the curvilinear relationship between the CW, CH, and CS at the same time	Variety	Type IV (a_1_ *, −a_2_, −a_3_ *, a_4_ *)	These four anchors have had significantly positive values on a_2_ and a_4_ and express the convex surface
Entrepreneurship	Type III (a_1_ *, a_2_, −a_3_, −a_4_ *)		Technical Competence	Type V (a_1_ *, a_2_, −a_3_, a_4_ *)	
Identity	Type II (a_1_ ***, −a_2_, a_3_, −a_4_)	These three anchors of its personnel have had significantly positive value a_1_ relationship between the CW, CH, and CS	Entrepreneurship	Type VI (a_1_, a_2_ *, a_3_, a_4_)	
Challenge	Type III (a_1_ ***, a_2_, −a_3_, −a_4_)		Managerial Competence	Type VII (−a_1_, a_2_ **, a_3_, a_4_)	
Lifestyle	Type III (a_1_ *, a_2_, −a_3_, −a_4_)		Challenge	Type IV (a_1_ *, −a_2_, −a_3_, a_4_)	These three anchors only have linear positive slop value a_1_ relationship between the CW, CH, and CS
Technical Competence	Type I (a_1_ ***, −a_2_ *, −a_3_ *, −a_4_ *)	These eight anchors have a significant curvilinear relationship between the CW, CH, and CS	Service	Type IV (a_1_ **, −a_2_, −a_3_, a_4_)	
Creativity	Type I (a_1_ ***, −a_2_ *, −a_3_, −a_4_ *)		Creativity	Type VI (a_1_ **, −a_2_, a_3_, a_4_)	
Service	Type I (a_1_ ***, −a_2_ *, −a_3_, −a_4_ *)		Geographical Security	Type IV (a_1_, −a_2_, −a_3_ ***, a_4_)	This anchor only has linear negative slop relationship between the CW, CH, and CS
Learning Motivation	Type I (a_1_ ***, −a_2_, −a_3_, -a_4_ **)		Identity	Type V (a_1_, a_2_, −a_3_, a_4_)	The positive value a_2_ of these two anchors have had marginally significant relationship between the CW, CH, and CS
Geographical Security	Type I (a_1_ ***, −a_2_, −a_3_, −a_4_ *)		Autonomy	Type VII (a_1_, a_2_, a_3_, a_4_)	
Organizational Stability	Type II (a_1_ ***, −a_2_, a_3_, −a_4_ *)		Learning Motivation	Type IV (a_1_, −a_2_, −a_3_, a_4_)	The positive value a_1_ of these two anchors also has had marginally significant relationship between the CW, CH, and CS and expresses the linear slope shape
Variety	Type II (a_1_ ***, −a_2_, a_3_, −a_4_ ***)		Lifestyle	Type IV (a_1_, −a_2_, −a_3_, a_4_)	
Autonomy	Type II (a_1_ ***, a_2_, a_3_, −a_4_ ***)		Organizational Stability	Type VI (a_1_, −a_2_, a_3_, a_4_)	This anchor does not have significant relationship between the CW, CH, and CS

***: *p* < 0.001 (t > 3.29); **: *p* < 0.01 (t > 2.58); *: *p* < 0.05 (t > 1.96).

## Data Availability

Data are available upon request to the contact author.

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
