# Peer review of "The Relationship between Discrepancies in Career Anchors of Information Technology Personnel and Career Satisfaction"

_behavsci, 2023, doi:10.3390/bs13090785_

Round 1
Reviewer 1 Report (Previous Reviewer 1)
The researcher or researchers demonstrated a good ability to modify the research, which makes it now ready for publication
Author Response
The researcher or researchers demonstrated a good ability to modify the research, which makes it now ready for publication.
Authors' response: Thank you for your comment.
Reviewer 2 Report (Previous Reviewer 2)
While going through all the suggested changes, I observed that this time the authors have revised the manuscript thoroughly. All changes have been incorporated. All suggestions have been followed. From my standpoint, the manuscript is now in a better position to be published. The clear discussion and data analysis will be beneficial for the readers in anticipation.
Author Response
While going through all the suggested changes, I observed that this time the authors have revised the manuscript thoroughly. All changes have been incorporated. All suggestions have been followed. From my standpoint, the manuscript is now in a better position to be published. The clear discussion and data analysis will be beneficial for the readers in anticipation.
Authors' response: Thank you for your comment.
Reviewer 3 Report (Previous Reviewer 3)
The Authors took into account the recommendations and improved the article in terms of linguistic correctness, clarified the cultural differences between China and the UAE, added a survey in the appendix and improved the way of citing references. In addition, other changes (eg in the description of results and discussions) significantly improved significantly the quality of the article. It is now ready for publication.
Author Response
The Authors took into account the recommendations and improved the article in terms of linguistic correctness, clarified the cultural differences between China and the UAE, added a survey in the appendix and improved the way of citing references. In addition, other changes (eg in the description of results and discussions) significantly improved significantly the quality of the article. It is now ready for publication.
Authors' response: Thank you for your comment.
Reviewer 4 Report (New Reviewer)
This paper aims at the relationship between Information Technology Personnel 2 Career Anchors' Discrepancy and Career Satisfaction. Based on the cultural society of Mainland China (the PRC) and United Arab Emirates (the UAE). this paper explains how information technology personnel can improve career satisfaction from the perspective of individuals and organizations. The research questions and results have strong theoretical and practical value. However, the paper still has the following problems, and it is hoped that the authors can make further improvements.
1. In abstract, the description of the conclusion of the paper is too complex and needs to be focused and more concise.
2. There are few literatures in this paper in recent three years, so it is suggested to review the latest research results, such as: https://doi.org/10.56578/jimd010206; https://doi.org/10.56578/esm010104; https://doi.org/10.3390/bs13030229
3. In the equation Z = b0 + b1 * X + b2 * Y + b3 * X2 + b4 * X * Y + b5 * Y2 + e; There are too many values to explain, change it to a table to look clearer
4. In Table1. Age(4), why is the column not rounded?
5. In "3.3. Reliability and Validity", why are the reasons for the method selection of PLS divided into two paragraphs?
Quality of English Language is ok
Author Response
This paper aims at the relationship between Information Technology Personnel 2 Career Anchors' Discrepancy and Career Satisfaction. Based on the cultural society of Mainland China (the PRC) and United Arab Emirates (the UAE). this paper explains how information technology personnel can improve career satisfaction from the perspective of individuals and organizations. The research questions and results have strong theoretical and practical value. However, the paper still has the following problems, and it is hoped that the authors can make further improvements.
Authors' response: Thank you for your comment.
- In abstract, the description of the conclusion of the paper is too complex and needs to be focused and more concise.
Authors' response: Thank you for your comment.
The content of the abstract is revised according to the comments of reviewer 1. The new version has deleted the final sentence, and it is more focused and concise now (p. 1). Hopefully the description is now acceptable.
- There are few literatures in this paper in recent three years, so it is suggested to review the latest research results, such as: https://doi.org/10.56578/jimd010206; https://doi.org/10.56578/esm010104; https://doi.org/10.3390/bs13030229
Authors' response: Thank you for your comment.
The three articles have cited in the manuscript, and hopefully the description is now acceptable.
- In the equation Z = b0 + b1*X + b2*Y + b3*X2 + b4*X*Y + b5*Y2 + e; There are too many values to explain, change it to a table to look clearer
Authors' response: Thank you for your comment.
The meaning of these values has added in the Table 4 (p. 11), and hopefully the description is now acceptable.
- In Table1. Age (4), why is the column not rounded?
Authors' response: Thank you for your comment.
The value has rounded in the new version (p. 6), and hopefully the description is now acceptable.
- In "3.3. Reliability and Validity", why are the reasons for the method selection of PLS divided into two paragraphs?
Authors' response: Thank you for your comment.
These two paragraphs have merge in the new version (p. 7), and hopefully the description is now acceptable.
Round 2
Reviewer 4 Report (New Reviewer)
The authors have revised paper well according to my comments
Quality of English Language is ok
This manuscript is a resubmission of an earlier submission. The following is a list of the peer review reports and author responses from that submission.
Round 1
Reviewer 1 Report
The Relationship between ITP Career Anchors' Discrepancy and Career Satisfaction
1- In the title: it is preferable to use words in full, and not to use abbreviations, especially since you have used abbreviations for all factors in the abstract.
2- The abstract section:
- The abstract should not contain abbreviations. At least When the variable is mentioned for the first time in the paper.
- The information that was included about the conditions of the sample and population, how to determine the sample size, and what was excluded from the responses is not completely clear. More clarification must be worked on, to confirm the possibility of generalizing the results to the community.
- The same observation applies to the part of the paper where the sample and the population fall, p5- line 228 in (Survey Administration section).
- In the abstract, the sector in which the study was conducted should be mentioned and time.
- In abstract the finding No (1) (The ITP in the PRC can be divided into Type I, Type II, and 18 Type III three categories, and there are totally different four categories (Type IV, Type V, Type VI, 19 and Type VII) of the ITP in the UAE). Is this being the way to show the result in abstract. it is first time for me to see this
- The abstract must be concise, clear, with a clear and objective view of the main concepts in the paper.
- The abstract must show the scientific novelty and the practical significance of the results. There is no need for abstract if it does not satisfy the reader's knowledge in a concise manner.
3- In introduction section:
- line 37 please use both full word and abbreviation When the variable is mentioned for the first time in the paper, later you can use the shortcut.
- The Same comment for all the variables in introduction section, When the variables is mentioned for the first time in the paper, please use both full word and abbreviation.
- In the introduction needs to explain and emphasize the limitations of the previous research findings and the significance of the current study.
- From page 1 to page 5 a lot of references were cited in the same line as line number (36, 42, 47, 51, 64, 107, 143, 200), I suggest reducing the number to as few as possible and deleting old references.
4- In Theoretical Backgrounds section
- Line 107, which is the only justification for cited many citations due to the researcher's desire to mention all the previous studies that touched on the topic (ITP), but it also opens the door to the question about the research gap or the contribution of the paper, as this was not clearly mentioned in the introduction or the abstract, even it was detailed in contribution section on page 17.
- The methodology used is a comparative method however, Despite the collection of various information about the sample, such as demographic and organizational factors, the theoretical framework did not cover these factors. This is an issue that must be seriously modified if one of the aims of the paper is to highlight these differences. Or add an explanation of why this information was collected and analyzed (demographic factors). For example, to describe the characteristics of the sample or to emphasize diversity.
5- Methodology section:
- in line 232 you said (the samples representativeness of this study is assured). Can you please explain how?
- And, how you calculate the sample and the count of population in (PRC and UAE), much information missed in this section.
- In general, no adequate explanation has been given for table number one.
- In line 246 you said that
(The Simplified Chinese is the official language in the PRC; thus the questionnaire items were translated into Simplified Chinese. Two professional translators independently translated the English questions into Chinese version)
My question is:
Why did you not do the same with the UAE. Although you indicated that (English is the second official language in the UAE), or rather, you did not translate the questionnaire into Arabic and vice versa.
At least give an explanation why.
When carrying out a comparison process, the conditions should be similar as possible, especially in the use of indicators (the language of the questionnaire).
6- in conclusion, please comment on the methods and results. If there are any objective errors, or if the conclusions are not supported, you should detail your concerns.
7- Any reference that goes back more than 30 years is considered outdated. Also, the relationship of the reference to the title of the study must be considered.
for example:
n T. J. Delong, “Reexamining the career anchor model personal,” Personnel, vol. 59, no. 3, pp. 50-63, 1982.
n J. R. Edwards, “The study of congruence in organizational behavior research: Critique and a proposed alternative,” Organizational Behavior Human Decision Processes, vol. 58, no. 1, pp. 51-100, 1994.
n J. R. Edwards, and M. E. Parry, “On the use of polynomial regression equations as an alternative to difference scores in organizational research,” Academy of Management Journal, vol. 36, no. 6, pp. 1577-1613, 1993.
n C. R. Fronell, A second generation of multivariate analysis methods. New York: Praeger. and Fred L. Bookstein,”Two Structural Equation Models: LISREL and PLS Applied to Consumer Exit-Voice Theory,” Journal of Marketing Research, vol. 19, pp. 440-452, 1982.
n C. Fornell, and D. F. Larcker, “Structural equation model with unobservable variables and measurement error: Algebra and statistics,” Journal of Marketing Research, vol. 18, no. 3, pp. 382-389, 1981.
n J. H. Greenhaus, S. Parasuraman, and W. M. Wormley, “Race effects of organizational experience, job performance evaluation, and career outcomes,” Academy of Management Journal, vol. 53, no. 1, pp. 64-96, 1990.
n M. Igbaria, and J. J. Baroudi, “A short-form measure of career orientations: a psychometric evaluation,” Journal of Management Information Systems, vol. 10, no. 2, pp. 131-154, 1993.
n A. C. Michalos, “Multiple discrepancies theory (MDT),” Social Indicators Research, vol. 16, no. 4, pp. 347-413, 1985.
n A. C. Michalos, “Job satisfaction, marital satisfaction, and the quality of life: A review and a preview,” in F. M. Andrews (ed.) Research on the Quality of Life (pp. 57-83). Ann Arbor, HI: University of Michigan Institute for Social Research, 1986.
n L. W. Porter, “A study of perceived need satisfaction in bottom and mid-management jobs,” Journal of Applied Psychology, vol. 45, pp. 1-10, 1961.
n R. W. Rice, D. B. McFarlin, and D. E. Bennett, “Standards of comparison and job satisfaction,” Journal of Applied Psychology, vol. 74, no. 4, pp. 591-598, 1989.
n E. H. Schein, “How career anchors hold executives to their career paths,” Personnel, vol. 52, no. 3, pp. 11-24, 1975.
n E. H. Schein, Career Dynamics: Matching Individual and Organizational Needs. Addision-Wesley, MA, 1978.
n 69. E. H. Schein, “Culture as an environmental context for careers,” Journal of Occupational Behavior, vol. 5, no. 1, pp. 71-81, 1984.
n 70. E.H. Schein, “A critical look at current career development theory and research,” In D. T. Hall, (Ed.), Career Development in Organizations. San Francisco: Jossey-Bass, pp. 310-331, 1986.
See:
http://dx.doi.org/10.4018/978-1-4666-0020-1.ch015
https://doi.org/10.1080/07408170500232784
https://doi.org/10.5465/256156
please consider the following note :
- Many references are very old.
- Many references not related to the topic under study.
Reviewer 2 Report
This is an interesting area of research and the results are significant. However, the following few suggestions are forwarded for the improvement of the manuscript.
1) ITP should be spelt out first and then may be used later. Also, spelling it out in the title may be better as I didn’t see it as a standard abbreviation on the Internet.
2) There are several abbreviations used in the introduction section and other parts of the manuscript which may confuse the readers. The authors are suggested to spell them out.
3) Could be better if the discussion is provided about the population, and respondents of the study, spelling out the abbreviations, sample size and techniques, and sample type;
4) Could be better if a discussion was provided about the data collection instruments, procedure and analysis techniques.
5) Could be better if the inclusion and exclusion criteria for the study participants are provided;
6) The 5age of PRC and UAE are confusing. Needs further explain how it was calculated;
7) I think there is a serious problem with the data, the total received responses are 369, while based on gender distribution, the total is 309. If missing values are removed, it becomes 308; there is a need to be justified where the rest of the responses are;
8) The same problem is with the age-based data distribution; The authors may calculate the total; in conclusion, the data distribution based on demographics is incorrect.
9) All the result section is seriously problematic and needs to be revisited.
10) Several grammatical errors are there in the whole manuscript;
11) Data instrument is missing; could be better to provide;
Reviewer 3 Report
The article presents interesting research results of theoretical and practical importance and has the potential to be a good scientific article, but after correcting the sometimes incomprehensible language of the text and non-transparent style, which definitely makes it difficult for the reader to understand the article. Extensive editing of English language and style is absolutely necessary.
Below there are some examples of incomprehensible sentences and language errors:- lines 62-63 there is no logical connection between the subordinate and superior clause: ‘Although a number of scholars have studied the internal career of employees, and the important skills of ITP, effective ITP should enhance their skills through their career path [16][31][48], the career aspirations and their career success [27][16]’,
- lines 38-39 Therefore, the organization can retain qualified ITP to improve its core capability, because Information systems (IS) will be a useful approach ITP (????) to increase the core capability of organizations, and qualified ITP can maintain it well- unfinished sentence lines 48-49: : Because the organizational structure and social transitions have influenced many aspects of professionals' career behavior, an effective professional who has to be equipped to cope with any problem and accomplish unique outcomes with limited resources within critical time constraints;
- the abbreviation CH translated as ‘CAREER HAVE’ ???
- line 97: incomprehensible sentence ‘Then, the ITP is different from other occupations to achieve the key objective of the current study’,
- lines 76-77 ‘how the gap further effects on their CS’ (to have effect on sth’ or ' to effect sth’, "to affect sth'.)
- line 226: lack of the verb: All respondents' consent for using survey to collect data.
- line 150: 'the gap between what workers have supported by their organization and what they want'
- line 83: there is no logical connection between the subordinate and superior clause: ‘Because culture is another important factor influencing the career anchors of the ITP, while previous studies have established the CW only’,
- line 123: ‘Although many scholars have studied the career anchors of ITP as well as above literature’ Do the authors mean that many scholars have studied above literature ? it doesn't make much sense.
In my opinion, the authors use too many abbreviations in the text. At the same time, they did not explain the abbreviation ITP anywhere.In the theoretical part of the article (background), the authors could explain in more detail the cultural differences between China and the UAE. Admittedly, in section 4.4. (Compare the Categories' Result of the ITP in the PRC and UAE Two Different Cultural Societies) they refer to these differences as part of the discussion, but they should describe cultural differences beforehand in a structured way, based on the literature review, and not just only as part of a discussion.
The authors clearly define the research questions. However, they could also consider formulating hypotheses. They could also consider including a survey (in appendix).
According to the journal’s instruction for the authors: ‘references must be numbered in order of appearance in the text (including table captions and figure legends) and listed individually at the end of the manuscript’. Reference in the article are numbered alphabetically.